# Effects of Microstructure and Chemical Composition on the Visual Characteristics of Flattened Bamboo Board

Lisheng Chen [1,†], Caiping Lian [2,3,†], Meiling Chen [2] and Zhihui Wu [2,*]

1   School of Materials Science and Engineering, Guizhou Minzu University, Guiyang 550025, China; chenlishengv5@163.com
2   College of Furnishing and Idustrial Design, Nanjing Forestry University, Nanjing 210037, China; liancaiping@njfu.edu.cn (C.L.); meiling_chen@njfu.edu.cn (M.C.)
3   School of Design, Fujian University of Technology, Fuzhou 350118, China
*   Correspondence: wzh550@njfu.edu.cn
†   These authors contributed equally to this work.

**Abstract:** Flattened bamboo board is a new type of bamboo-based panel with various colors that maintains the natural texture of bamboo, and is gradually being used in indoor home decoration. Revealing the influence mechanism on the visual effect of flattened bamboo boards is the key to improving the processing of such boards for household materials. This study employed visual physical quantity measurement methods, field emission scanning electron microscopy, FTIR spectroscopy, and XPS to investigate the visual physical quantities, morphology, and chemical composition of flattened bamboo boards. The results showed that compared with the control samples, the bamboo outer layer boards were dark brown, with the largest $\Delta E^*$ (38.55), while the outer boards were reddish-brown, with the largest $a^*$ (8.82). The inner boards were yellow-red and showed a lower $\Delta E^*$ (6.55). Due to the elevated density, abundant inclusion, and wax, the bamboo outer layer board exhibited the highest glossiness and darkest color, followed by the outer board and the inner board. The FTIR spectroscopy revealed that hemicellulose decomposed, and the relative content of lignin increased, leading to color changes in the flattened bamboo boards. The bamboo outer layer board was the darkest due to changes in C=C bonds at 1600 cm$^{-1}$ and 1509 cm$^{-1}$. The surface color of the outer board was mainly red, which may be caused by C–O bonds at 1239 cm$^{-1}$. The surface of the inner board was mainly yellow, which may be caused by the C–H stretching vibration of lignin at 1108 cm$^{-1}$. XPS analysis showed that the proportion of C1 and O1 increased, while C2, C3, and O2 decreased, indicating that hemicellulose degraded at high temperatures, which increased the relative lignin content. Changes in the relative content of oxygen-containing functional groups and SiO$_2$ in the flattened bamboo board were important factors responsible for the change in visual physical quantities.

**Keywords:** flattened bamboo boards; visual physical quantities; morphology; chemical composition; FTIR; XPS





## 1. Introduction

Flattened bamboo board is produced by softening bamboo tubes at high temperatures and then pressing them into sheet plates [1]. In recent years, flattened bamboo board has been used in interior decoration and furniture due to its high mechanical properties and the natural texture of bamboo [2]. Flattened bamboo board is available in a variety of colors, and the bamboo outer layer board can replace rosewood to impart a decorative effect to furniture. After being subjected to high-temperature softening and pressure flattening, the visual physical quantities, i.e., color, of the flattened bamboo board change significantly. However, research on flattened bamboo boards has mainly focused on softening and flattening technologies and mechanisms, as well as the performance and surface visual characteristics [3–10], while the factors influencing its visual features have not been clarified.

Changes in the surface color of most materials mainly occur due to the decomposition of chemical components or the generation of colored groups or chromatic groups. High-temperature softening mainly occurs due to the viscoelasticity imparted by lignin, cellulose, and hemicellulose above their glass transition temperatures, thereby realizing non-crack flattening. It is unclear whether high-temperature softening and flattening change visual physical quantities such as color or glossiness or whether the chemical components of the cell wall are pyrolyzed during softening. It is also unknown which groups decompose or form at high temperatures and cause color changes in flattened bamboo board or different visual physical quantities between different flattened bamboo boards. At present, no studies have analyzed these problems, even though they directly affect the processing technology of flattened bamboo boards, and thus the visual effect of flattened bamboo boards.

To reveal the effects of high-temperature softening and flattening on visual physical quantities of flattened bamboo boards, a spectrophotometer was used to determine the color of flattened bamboo boards and control samples. Changes in microstructure were also determined using field emission scanning electron microscopy (SEM) to investigate its effects on the color of flattened bamboo boards. FTIR and XPS were employed to analyze the chemical composition of flattened bamboo boards. Through unveiling the influence of microstructure and chemical composition on the visual characteristics of flattened bamboo boards, this study offers essential theoretical support for enhancing the value-added potential of bamboo products. This is of great significance to the development of the bamboo industry.

## 2. Materials and Methods

### 2.1. Materials

The sample materials were the fourth section of bamboo tubes of three-year-old *Phyllostachys edulis* taken from Nanping, Fujian. Part of this sample material was used for the preparation of flattened bamboo boards, and the other part of this sample material was used for preparing the control samples. Flattened bamboo board was prepared by Fujian Longzhu Group Co., Ltd. (Nanping, China) using a non-notched, high-temperature softening and flattening process with a softening temperature of 175–180 °C. Flattened bamboo boards were divided into three categories: bamboo outer layer board F1 (keeping the bamboo green and yellow), outer board F2 (cut bamboo green surface 1 mm thick), and inner board F3 (cut bamboo yellow surface 1 mm thick). The specific production process and illustration regarding these were mentioned in our previous study [10]. Each type of flattened bamboo board sample had dimensions of 60 mm × 40 mm × 5 mm (length × width × thickness) measured with a vernier caliper. The control samples were used to assess alterations in the visual characteristics, microstructure, and chemical composition of bamboo before and after the process of high-temperature flattening. The difference between the flattened bamboo boards and the control samples was that the control samples were not treated with high-temperature softening and pressure flattening. The control samples were also divided into three categories: the bamboo outer layer board Ct1 (keeping the bamboo green and yellow), outer board Ct2 (cut bamboo green surface 1 mm thick), and inner board Ct3 (cut bamboo yellow surface 1 mm thick). Five pieces of both flattened bamboo boards and control samples were included in each group, and the mean value of the final results was taken. The flattened bamboo boards and control samples are shown in Figure 1.

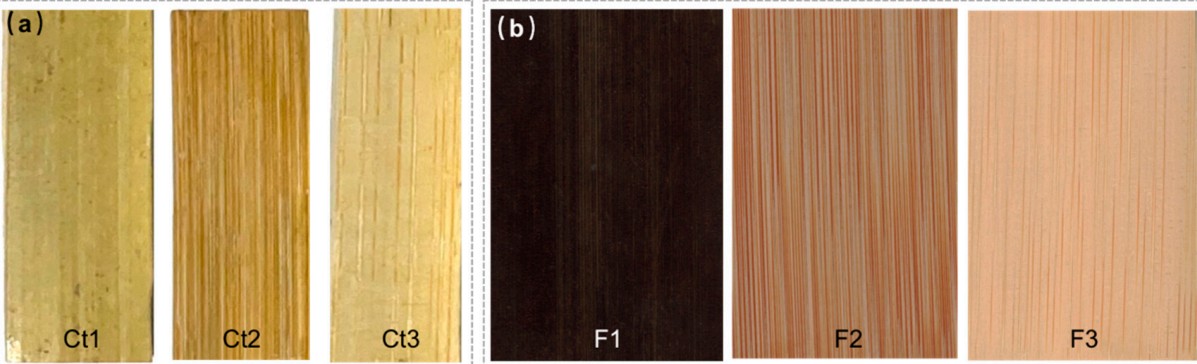

**Figure 1.** The appearance of the flattened bamboo boards and natural bamboo. (**a**) The control samples included bamboo outer layer Ct1, the outer board Ct2, and the inner board Ct3. (**b**) The flattened bamboo boards included bamboo outer layer board F1, the outer board F2, and the inner board F3.

*2.2. Colorimetric Analyses*

Surface color determination of the flattened bamboo boards and control samples was carried out according to the chromaticity space CIE coordinates *L*a*b** [11,12]. A spectrophotometer (X-Rite PANTONE Series spectrophotometer, RM200, USA) was used to determine the color of the three groups of flattened bamboo boards and control samples. Six points were measured on the surface of each sample, and the average value of the six points was used as the color index of each sample. The color meter used a D65 standard light source, the observation angle was set at 10°, and the optical aperture was fixed at 8.0 mm. The CIE *L*a*b** 1976 system consisted of three parameters: *L** is the brightness index, where larger values indicate a higher brightness; *a** is the red and green product index, where negative values indicate green, and positive values indicate red; *b** is the blue-yellow index, where negative values indicate blue, and positive values indicate yellow. The color change of the samples before and after heat treatment was expressed by the total color difference index Δ*E**:

$$\Delta E^* = \sqrt{\Delta L^2 + \Delta a^2 + \Delta b^2} \tag{1}$$

where Δ*E** represents the total color difference; Δ*L** represents the change in the brightness index of the test material before and after heat treatment; Δ*a** represents the change in the red-green index before and after heat treatment; and Δ*b** represents the change in the yellow-blue index before and after heat treatment.

*2.3. Glossiness Measurements*

Glossiness is a crucial parameter reflecting the ability to reflect light on the surface of a material. The surface glossiness of bamboo blocks and the control samples was measured with a gloss meter (HG268, Shenzhen 3nh Technology Co., Ltd., Shenzhen, China) according to the national standard ISO 2014 [13]. The selected optical geometric condition was an incidence angle of 60°, and the glossiness was determined based on a specular glossiness value of 100, which represents a perfect, highly-polished surface under the same lighting and viewing conditions. The incidence direction of the light source was selected in two directions: parallel to the processional texture ($G_{ZL}$) and perpendicular to the texture ($G_{ZT}$). Three measurement points were selected for testing each sample, and the average value was considered as the test result.

*2.4. SEM Analysis*

SEM samples were prepared following the procedure outlined by Lian et al. [14]. There were three groups of prepared SEM samples of flattened bamboo boards with corresponding control samples, which were fixed on a sample holder. Subsequently, the sample surface was sprayed with platinum powder for 90 s to coat the bamboo block with

an 8 mm thick layer of platinum. The platinum-sprayed samples were then placed in a field emission scanning electron microscope (Quanta 200, FEI Company, Hillsboro, OR, USA) for morphology observations using an acceleration voltage of 7–10 kV.

### 2.5. FTIR Spectroscopy

Three groups of flattened bamboo boards and control samples were prepared into blocks with dimensions of 30 mm × 30 mm × 5 mm (length × width × thickness). These blocks were subsequently dried to absolute dryness in an oven before being subjected to infrared spectrometry (VERTEX 80V, Bruker, Bruck, Germany). The infrared spectrometer utilized a spectral wavenumber range of 4000–500 cm$^{-1}$, a resolution of 4 cm$^{-1}$, and a scanning frequency of 64.

### 2.6. XPS Analysis

In accordance with the requirements of XPS, the flattened bamboo boards and control samples were prepared into flat surface block samples measuring 1 mm × 3 mm × 3 mm. A total of six groups of samples were prepared and dried for testing. During the preparation process, care was taken to avoid contact between hands and the sample surface to prevent contamination. The samples were placed in an X-ray photoelectron spectrometer (AXIS Ultra DLD, Kratos, Kawasaki-shi, Japan) with monochromatic Al K$\alpha$ as the target material (1486.6 eV). The sample was scanned in the analysis chamber, which maintainded a vacuum level exceeding $7 \times 10^{-8}$ Pa. The scanning power was 600 W and the scanning step was 0.1 eV. The sample analysis area was 700 μm × 300 μm. After the energy spectra were obtained, the relative concentrations of C and O atoms in the samples were analyzed according to Formula (2). XPS Peak Fit software (v4.1) was used for the peak-to-peak fitting of the spectrum, and the percentages of each chemical state of C, O, and Si elements were analyzed.

$$C = \frac{I_O/S_O}{I_C/S_C} \tag{2}$$

where $I_O$ represents the peak area of O atoms; $I_C$ represents the peak area of C atoms; $S_O$ represents the sensitivity factor of O atoms; and $S_C$ represents the sensitivity factor of C atoms.

### 2.7. Statistical Analysis

The color, glossiness, and chemical composition data of the flattened bamboo boards measured in this experiment were processed using Origin 2017 software.

## 3. Results and Discussion

### 3.1. Comparison of Visual Physical Quantities between Flattened Bamboo Boards and Control Samples

The effect of high-temperature softening treatment on the color of bamboo surface could be identified with the naked eye. Untreated bamboo exhibited a lighter coloration with a yellow-green hue. Flattened bamboo boards treated at 175–180 °C were darker, and the overall color of the outer board and the inner board was red, while that of the bamboo outer layer board was similar to that of rosewood (Figure 1).

The quantitative difference in color between flattened bamboo boards and control samples is presented in Table 1 and Figure 2. Generally, the brightness (*L*\*) and yellow-blue axis index (*b*\*) of flattened bamboo boards decreased after high-temperature softening treatment, while the red-green axis index (*a*\*) increased. Notably, the values of *a*\* and *b*\* remained positive, indicating an overall red-yellow appearance of the flattened bamboo boards, consistent with the color change of wood during heat treatment [15]. After treatment, the color indexes of the bamboo outer layer board were significantly different from those of the outer and inner boards (Δ*E*\* = 38.55), where *L*\* and *b*\* decreased by 51.2% and 64.4%, respectively, while *a*\* increased by 59.6%. The total color difference of the outer board was the smallest (Δ*E*\* = 4.88). Previous studies have demonstrated that *L*\* decreases at higher temperatures, while *a*\* and *b*\* initially increase and then decrease [16,17]. In our

study, *L** and *b** of the flattened bamboo boards decreased after the bamboo was flattened with a high temperature and pressure, while *a** increased.

**Table 1.** The results of visual physical quantities of flattened bamboo boards and control samples.

| Sample | *L** | *a** | *b** | Δ*E** | $G_{ZL}$ (%) | $G_{ZT}$ (%) |
|---|---|---|---|---|---|---|
| F1 | 31.82 | 4.23 | 10.54 | | 5.99 | 4.77 |
| Ct1 | 65.21 | 1.71 | 29.65 | 38.55 | 5.78 | 4.52 |
| F2 | 63.67 | 8.82 | 27.23 | | 4.17 | 2.95 |
| Ct2 | 65.82 | 8.60 | 31.61 | 4.88 | 4.05 | 2.78 |
| F3 | 73.21 | 6.91 | 25.82 | | 2.85 | 2.56 |
| Ct3 | 78.72 | 4.47 | 28.39 | 6.55 | 2.83 | 2.27 |

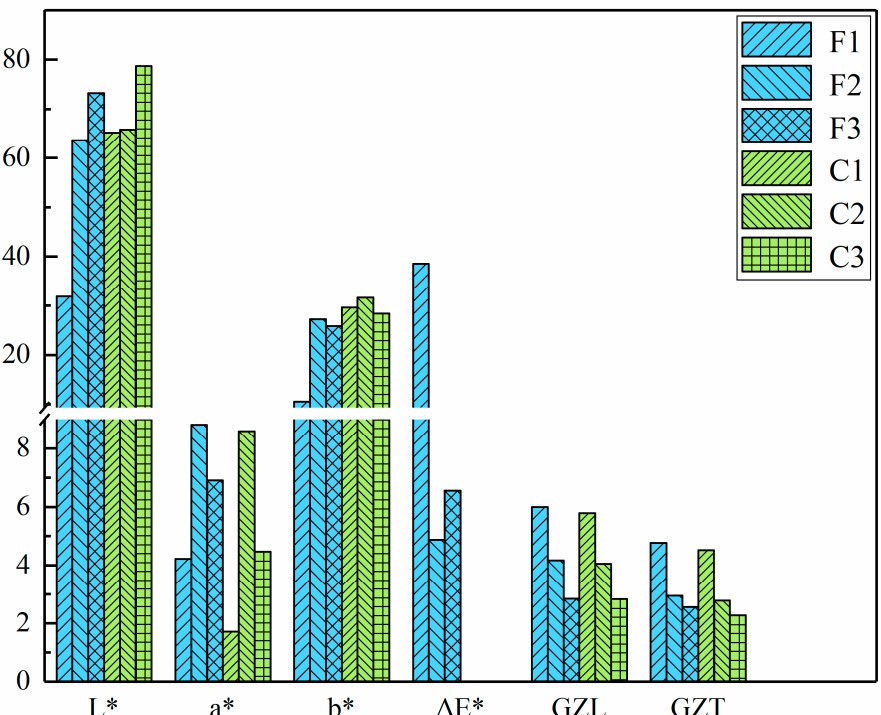

**Figure 2.** The change trend and difference of visual physical quantities between flattened bamboo boards and control samples.

In addition, the glossiness of the three types of flattened bamboo boards was higher than that of the control samples (Figure 2), indicating that softening and pressure treatment increased the glossiness of bamboo. However, both the flattened bamboo boards and the control samples exhibited similar trends, with the bamboo outer layer board consistently displaying the highest glossiness, followed by the outer board, and then the inner board.

### 3.2. Effects of Microstructure Differences on Visual Physical Quantities

The microstructural differences of the cross-section and longitudinal section (samples surface) and three kinds of flattened bamboo boards and control samples were compared to investigate the color and glossiness changes at the structural level. As shown in Figure 3, the cells of bamboo were compressed and deformed after high-temperature softening and high-pressure treatment. The vessels in the metaxylem changed from approximately circular to droplet-shaped, and the cell walls of the parenchyma cells changed from smooth ovals to an irregular shape, particularly on the outer surface of the bamboo outer layer board, where deformation was most obvious. As the cells were compressed, their cavities and pits became smaller, and the area occupied by the cell walls increased, resulting in denser flattened bamboo boards with a darker color, similar to the difference between

springwood and summerwood. These findings further indicate that the surface of the bamboo outer layer board underwent the greatest color change, which was consistent with changes in the total color difference index ($\Delta E^*$). In addition, some studies have shown that when the temperature is below 200 °C, the change in $\Delta E^*$ will be small due to the fact that the chemical composition will not be significantly degraded [18]. In this study, the bamboo materials were treated at 175–180 °C, so the $\Delta E^*$ of the outer and the inner board changed little, which was just consistent with the results of previous studies. However, due to the presence of a wax layer on the surface of the bamboo outer layer board, the decomposition of paraffin led to a significant deepening of the board surface, so the $\Delta E^*$ changed greatly.

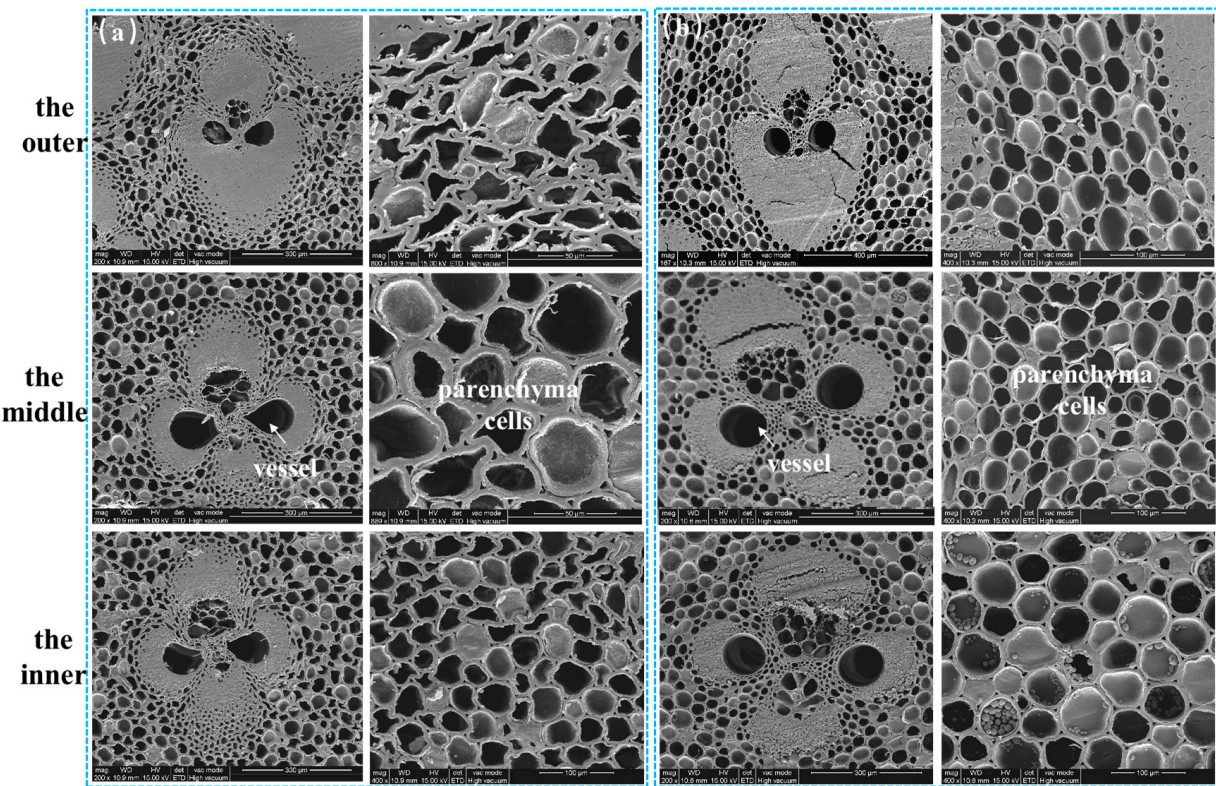

**Figure 3.** SEM images of the cross-section of the flattened bamboo boards (**a**) and control samples (**b**).

The stomata (Figure 4b,j), the only porous structures on the surface, were flattened and closed under high temperature and pressure, resulting in a non-porous wall on the surface of the bamboo outer layer board. There was a waxy layer on the surface of the bamboo outer layer board whose main component was paraffin wax [19], which was prone to darkening at high temperatures, thus causing the overall surface color of the bamboo outer layer board to darken. The outer primarily consisted of numerous fiber sheaths and basic tissues, while the inner was predominantly composed of parenchyma cells. According to a comparison of Figure 4c,f,i, the parenchyma cells of the outer were smaller than those of the inner, resulting in a higher density of the outer and thus a darker color.

In addition, the glossiness of the bamboo surface was related to its surface roughness, the structural characteristics of bamboo, extracts, and inclusions, and the incident light angle (reflection) [20]. As shown in Figure 4, the surface of the bamboo outer layer board was rich with inclusions (Figure 4(c1,c2)), while the outer board mainly contained dissolved starch residues (Figure 4(f1,f2)). There were fewer inclusions in the inner board with occasional starch paste (Figure 4(i1,i2)). Because the cell cavity filled with the inclusion was smaller, the light reflectance was increased, and the glossiness of the bamboo surface was improved. Based on previous findings, the entire surface of the bamboo outer layer board exhibited a wall-like structure. Because the proportion of cell walls in the outer was higher than that of the inner, the outer was denser than the inner; therefore, the bamboo outer

layer board exhibited the highest glossiness, followed by the outer board and then the inner board. This trend was also observed in the control samples. After high-temperature softening and pressure flattening treatment, the cells were compressed, forming a deformed structure with smaller cavities, thereby reducing the surface roughness of the flattened bamboo boards and enhancing the reflectivity of light, which may be a main reason why the glossiness of the flattened bamboo boards was higher than that of the control samples.

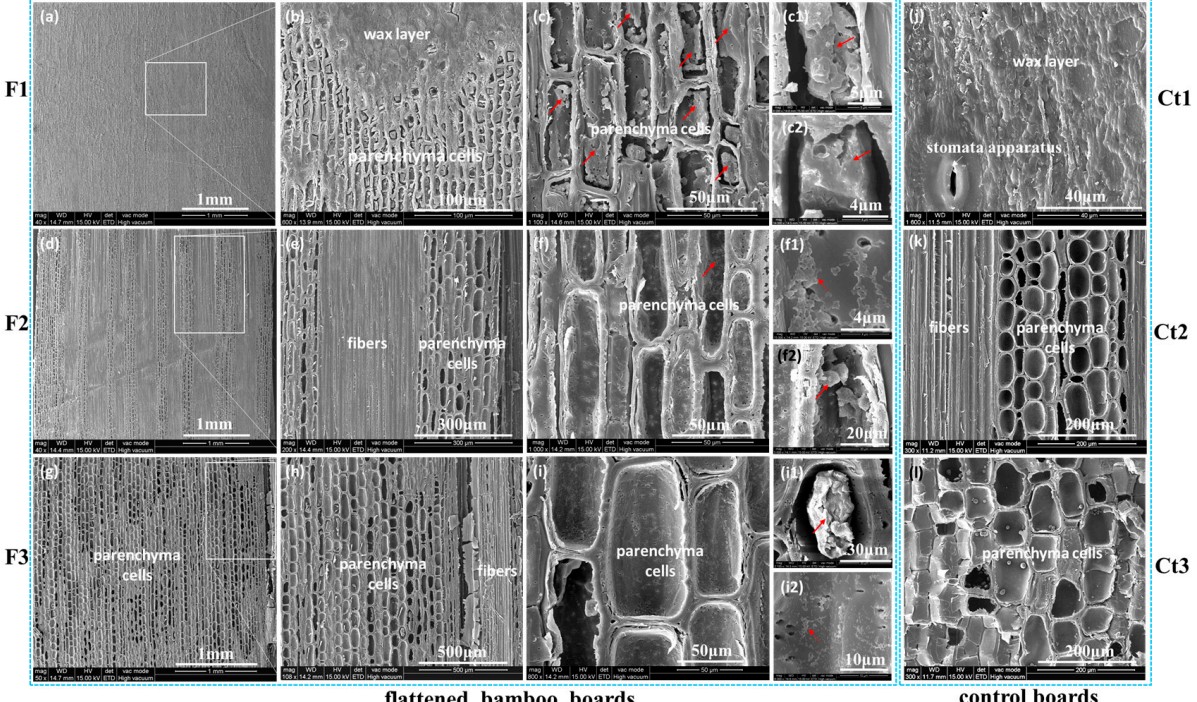

**Figure 4.** SEM images of the longitudinal section of the flattened bamboo boards and control samples. (**a**) The surface of the bamboo outer layer board; (**b**) enlarged image of (**a**); (**c**) the surface characteristics of parenchyma cells in the bamboo outer layer board; (**c1,c2**) inclusions in the parenchyma cell cavity of the bamboo outer layer board; (**d**) the surface of the outer board; (**e**) enlarged image of (**d**); (**f**) the surface characteristics of parenchyma cells in the outer board; (**f1,f2**) inclusions in the parenchyma cell cavity of the outer board; (**g**) the surface of the inner board; (**h**) enlarged image of (**g**); (**i**) the surface characteristics of parenchyma cells in the inner board; (**i1,i2**) inclusions in the parenchyma cell cavity of the inner board; and (**j–l**) the longitudinal characteristics of the control samples. The red arrows refer to inclusions.

### 3.3. Effects of Chemical Composition Differences on Visual Physical Quantities

### 3.3.1. FTIR Analysis

Bamboo, a natural, organic polymer, is mainly composed of three organic compounds: cellulose, hemicellulose, and lignin. The main infrared-sensitive group of cellulose is the hydroxyl group –OH; the infrared-sensitive group of hemicellulose is the carbonyl C=O; lignin has several more sensitive groups, including methoxy –$CH_3O$, hydroxyl –OH, carboxyl –COO, alkenes C=C, and aromatic rings. Structural changes in molecules can be predicted with analysis of the shifts in the direction, degree, and intensity of these characteristic groups [21]. In different radial parts of bamboo walls or during the softening treatment, these infrared-sensitive groups undergo different or varying degrees of change, leading to differences in the color of the material [22]. Infrared spectroscopy can be used to measure different processed or control materials to understand color differences caused by changes in the macromolecules of bamboo in different parts or under different conditions. The infrared spectra of the flattened bamboo boards and control materials are shown in Figures 5 and 6.

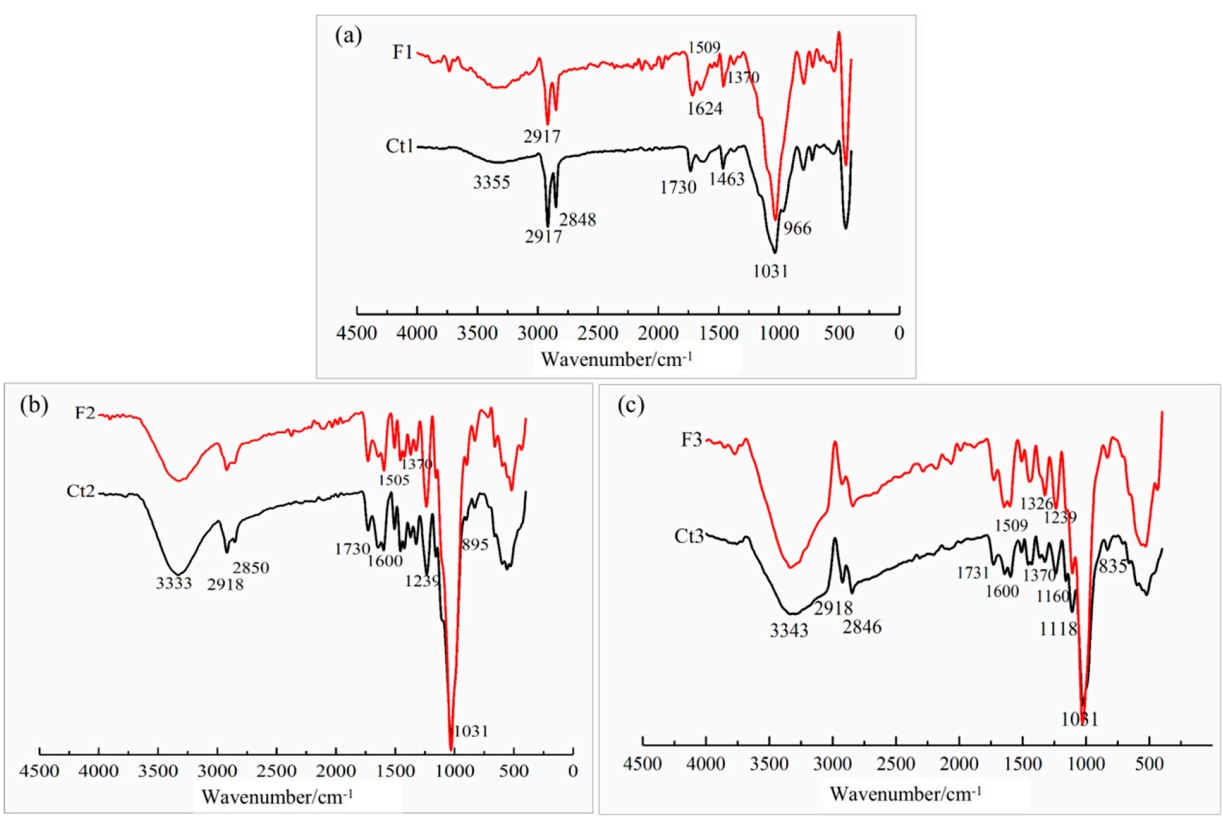

**Figure 5.** Comparison of FTIR spectra between flattened bamboo boards and control samples on radial surface. (**a**) Bamboo outer layer board; (**b**) the outer board; (**c**) the inner board.

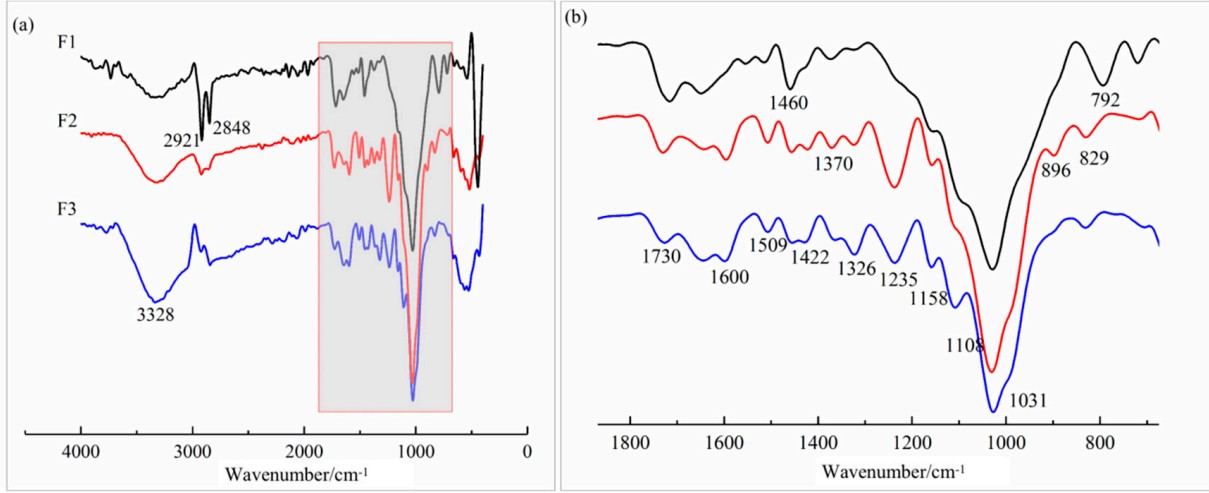

**Figure 6.** Comparison of FTIR spectra of three different flattened bamboo boards. (**a**) Three types of flattened bamboo boards; (**b**) enlarged image of (**a**).

The characteristic absorption peaks of cellulose appear at 2900 cm$^{-1}$, 1425 cm$^{-1}$, 1370 cm$^{-1}$, and 895 cm$^{-1}$. The carbonyl C=O stretching vibration absorption peak near 1730 cm$^{-1}$ is the characteristic peak of hemicellulose, which allows it to be distinguished from other components. The characteristic absorption bands of the lignin aromatic ring skeleton appeared near 1600 cm$^{-1}$ and 1505 cm$^{-1}$. By analyzing the infrared spectra of the radial surface of the flattened bamboo boards and control samples, and combining the characteristics of the absorption peaks [23–26], these characteristic absorption peaks of moso bamboo material are shown in Table 2.

**Table 2.** The typical absorbance peaks of moso bamboo material [23–26].

| Typical Absorbance Peaks (cm$^{-1}$) | Assignment |
| --- | --- |
| 3349 | –OH stretch (hydrogen-bonded) |
| 2917/2848 | C–H stretch (stretching of the methyl and methylene groups; hydrocarbon chains on cellulose) |
| 1730 | unconjugated C=O in xylan (hemicellulose) |
| 1600 | aromatic skeletal vibration (C=C) in lignin |
| 1505 | aromatic skeletal vibration (C=C) in lignin |
| 1460 | –CH$_3$ deformation in lignin and –CH$_2$ bending in xylan |
| 1370 | C–H deformation in cellulose and hemicellulose |
| 1239 | syringyl ring and C–O stretching in lignin and xylan |
| 1108 | C–H stretching (Guaiacyl and syringyl) |
| 1031 | C–O stretching (lignin, cellulose, and hemicelluloses) |

Research has shown that C–H bonds (2900 cm$^{-1}$) are more stable than other functional groups, so they are generally unaffected by heat treatment. Therefore, taking this absorption peak as the reference, the infrared spectra of the flattened bamboo board were compared with the reference material. Figure 5 shows that the differences in the infrared spectra of bamboo outer layer board, the outer board, and the inner board after high-temperature softening were similar. Compared with the control samples, the intensity of the absorption peak of C=O stretching vibration at 1730 cm$^{-1}$ in the flattened bamboo board decreased after high-temperature softening treatment (175 °C). This peak represented the non-conjugated carbonyl stretching peak of xylose in hemicellulose. Therefore, the decrease in intensity at 1730 cm$^{-1}$ may be due to the decomposition of hemicellulose above 160 °C [27]. The peak intensity at 1600 cm$^{-1}$ of the outer board and inner board increased. This is the signal of aromatic skeletal vibration (C=C) in lignin, and heat treatment increased the relative content of lignin [27]. However, the observed change in the peak intensity of the bamboo outer layer board at 1600 cm$^{-1}$ was minimal, and the control sample also lacked a prominent peak at this location. The peak intensity at 1505 cm$^{-1}$ (aromatic skeletal vibration in lignin) increased slightly because heat treatment broke the aliphatic side chain of lignin and caused lignin to undergo a condensation crosslinking reaction [28]. The absorption peak at 1370 cm$^{-1}$ was the C–H deformation in cellulose and hemicellulose. Due to the thermal decomposition of hemicellulose at high temperatures, its strength tended to decrease. The absorption peak at 1239 cm$^{-1}$ represented the syringyl ring and C–O stretching in lignin and xylan, and heat treatment increased this peak's intensity slightly. The obvious increase in the absorption peak intensity at 1031cm$^{-1}$ may indicate the production of new alcohols or esters. The FTIR spectra showed that high-temperature softening and pressure treatment degraded hemicellulose and increased the relative lignin content. This result was generally consistent with those of Meng et al. [24], who studied the surface chemical composition of heat-treated bamboo. In general, after high-temperature softening and pressure treatment, hemicellulose degraded and formed some oxygen-containing groups, such as carboxylic groups. Coupled with the increase in the relative content of lignin and the oxidation reaction of lignin, the color of bamboo boards eventually changed from yellow-green to reddish-brown or brown-dark, so the *L** and *b** decreased, while *a** increased.

Comparing and analyzing the infrared spectra of the surface of the bamboo outer layer board, the outer board, and the inner board (Figure 6) shows that the absorption peaks with significant differences were mainly located near 3000 cm$^{-1}$ and 800–1800 cm$^{-1}$. The spectrum of the bamboo outer layer board had strong absorption peaks at 2921 cm$^{-1}$ and 2848 cm$^{-1}$, while the inner board had the weakest absorption peak at 2921 cm$^{-1}$. The absorption peaks at these two sites indicated the stretching vibrations of the –CH$_3$, –CH$_2$, CH=O, and R$_3$C–H groups of cellulose [23]. There was no significant difference in peak strength at 1730 cm$^{-1}$ and 1370 cm$^{-1}$ between the three surfaces, indicating the same degree of hemicellulose decomposition for the three surfaces. The absorption peak intensity at 1600 cm$^{-1}$ of the bamboo outer layer board was the weakest, and the absorption peak at this position for the outer board was the highest, followed by the inner board. The absorption peak at 1509 cm$^{-1}$ was highest in the outer board, followed by the inner board,

and then the bamboo outer layer board. The absorption peak at 1460 cm$^{-1}$ was highest in the bamboo outer layer board, followed by the outer board, and then the inner board. This peak represents the –CH$_3$ deformation in lignin and –CH$_2$ bending in xylan. The absorption peak strength at 1239 cm$^{-1}$ of the bamboo outer layer board was weakest, and the absorption peak at this position in the outer board was higher than that of the inner board. The absorption peak at 1108 cm$^{-1}$ represents the C–H stretching vibration in the guaiacyl and syringyl of lignin, where the peak strength of the inner board was highest, followed by the outer board, and then the bamboo outer layer board. The spectrum of the bamboo outer layer board had strong absorption peaks at 1031 cm$^{-1}$. Aromatic rings, unsaturated double bonds, methoxy, carboxyl, and other oxygen-containing functional groups were important factors affecting the color index of bamboo [29,30]. Therefore, combined with the previous analysis of the color index, in addition to the influence of the wax layer, changes in the C=C bonds at 1600 cm$^{-1}$ and 1509 cm$^{-1}$ may have also been important factors, giving the bamboo outer layer board the darkest color. The outer board was mainly red, which may have been caused by its C–O content (1239 cm$^{-1}$). The inner board was mainly yellow, which may be caused by the C–H stretching vibration in the guaiacyl and syringyl of lignin at 1108 cm$^{-1}$.

3.3.2. XPS

FTIR can only perform semi-quantitative analysis of the chemical composition of bamboo materials, while XPS can quantitatively analyze the changes in elements. Therefore, combining XPS analysis, the impact of chemical composition on the visual characteristics of flattened bamboo boards can be better reflected. Similar to wood, bamboo is mainly composed of C, O, and H elements, but XPS cannot identify H atoms. The deconvoluted C 1s and O 1s XPS spectra are shown in Figures 7 and 8. There are three C 1s peaks, in which C1 represents saturated C atoms combined with C–H and C–C, which were mainly derived from lignin and extracts in the sample. C2 represents C atoms connected to a non-carbonyl O atom, derived from cellulose and hemicellulose. C3 represents a C atom connected to a carboxyl O atom or to two non-carboxyl O atoms derived from acetals in cellulose and hemicellulose and carboxyl groups in lignin. Two characteristic peaks, O1 and O2, representing the O atoms of OH–C=O and C–OH, appeared in the O 1s spectrum. The different chemical states and major binding energies of C and O elements in bamboo are listed in Table 3.

The degradation of wood materials can be judged using the change in the O/C ratio. The O/C values of natural wood cellulose and hemicellulose were 0.83 and 0.8, respectively, while that of lignin was 0.33 due to its high C content [31]. Table 4 shows that the O/C values of bamboo were all below 0.33, indicating that bamboo contained a relatively high lignin content. Among them, the O/C value of the bamboo outer layer board was the lowest, followed by the outer board, and the highest was the inner board. Because the wax layer on the surface of the bamboo outer layer board was mainly composed of alkanes, there was an extremely high C content and a lower O/C value. Due to the degradation of hemicellulose, there was an increase in the relative content of lignin, which reduced the O/C value. Except for the bamboo outer layer board, the O/C values of the other two flattened bamboo boards were lower than that of the control material, which was consistent with the FTIR spectra. However, the O/C value of the bamboo outer layer board after high-temperature softening treatment increased, possibly due to condensation reactions between C and O ions, which produced a large amount of O–C=O. This significantly increased the O1 content (Table 4), ultimately causing the deepest color change of the bamboo outer layer board.

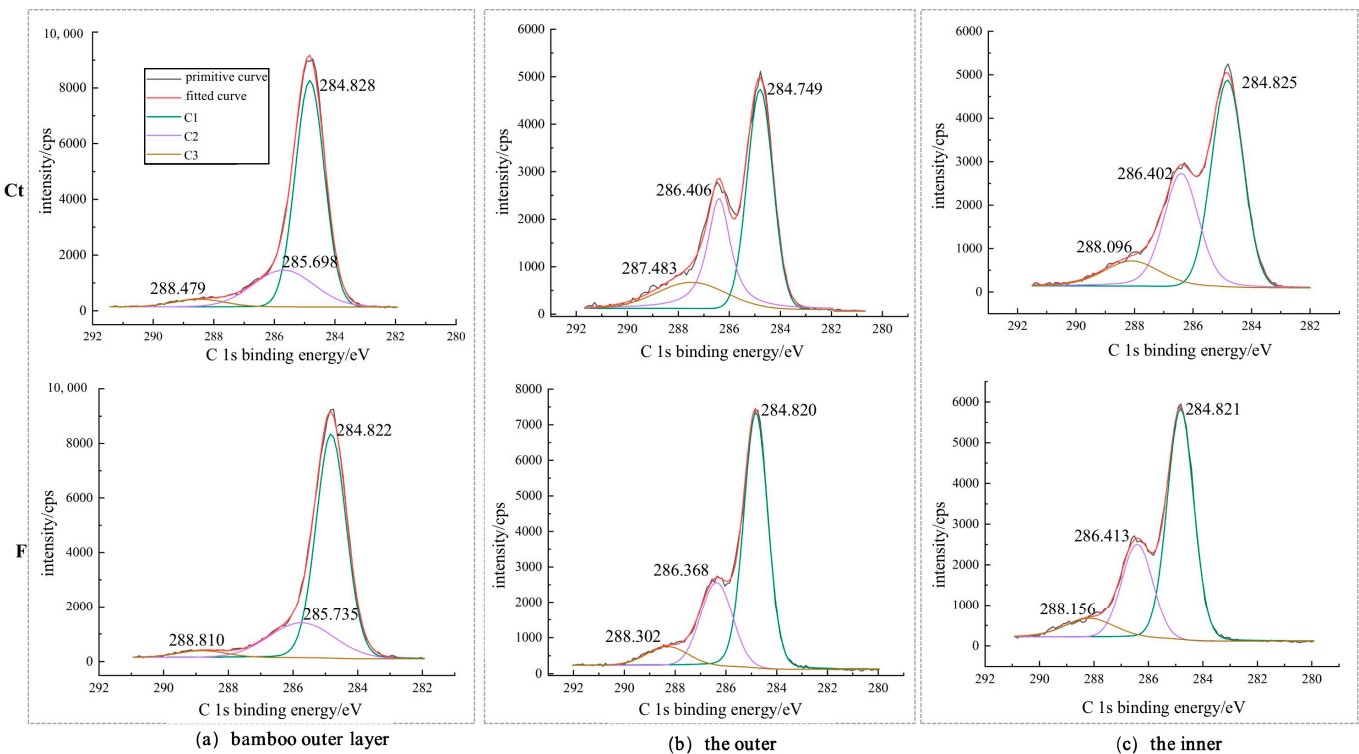

**Figure 7.** XPS curve of three chemical states of C element in flattened bamboo boards and control samples.

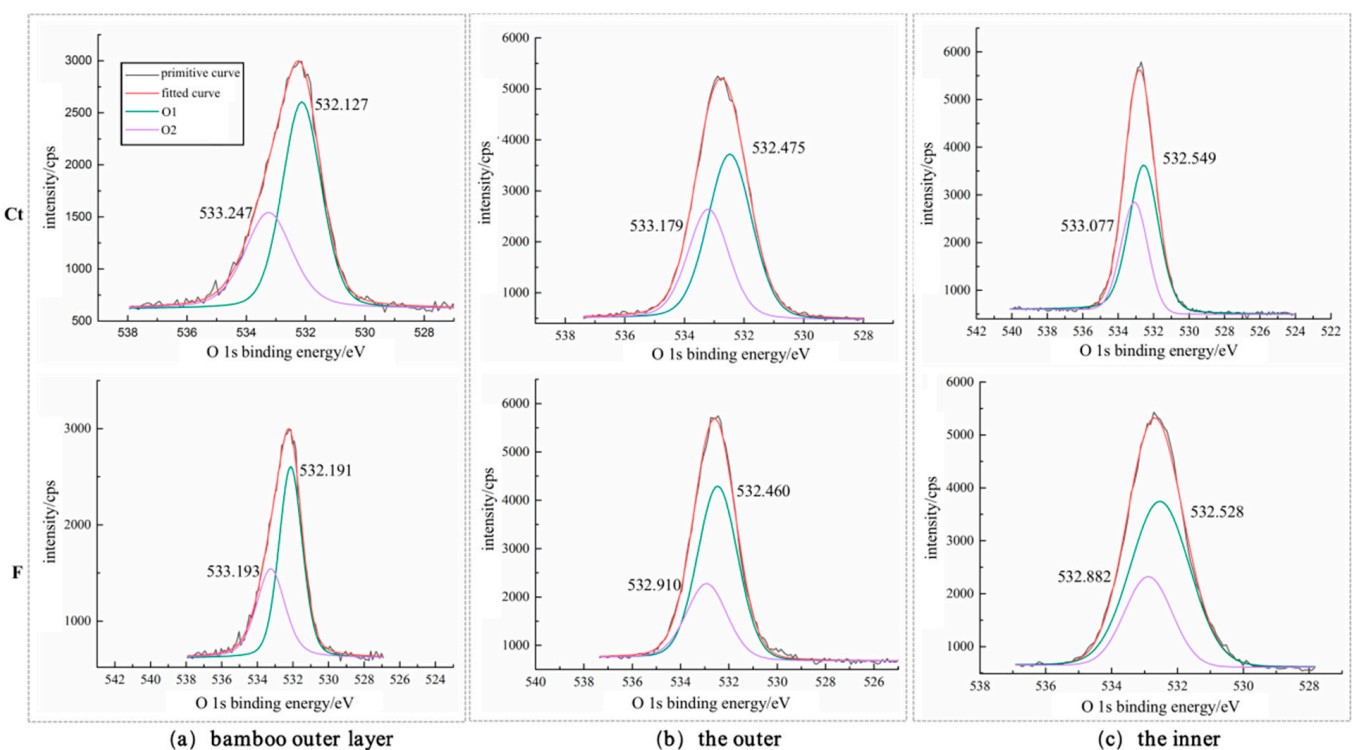

**Figure 8.** XPS curve of two chemical states of O element in flattened bamboo boards and control samples.

**Table 3.** Different chemical states and binding energy of C and O elements in bamboo.

| Element | Binding Energy/eV | Binding Form |
|---------|-------------------|--------------|
| C1 | 284.8 | C–H, C–C |
| C2 | 286.5 | C–OH, C–O–C |
| C3 | 288 | C=O, O–C–O |
| O1 | 532 | OH–C=O |
| O2 | 533 | C–OH |

**Table 4.** XPS analysis results of flattened bamboo boards and control samples.

| Sample | O/C | The Percentage of C Elements in Different Chemical States (%) | | | The Percentage of O Elements in Different Chemical States (%) | | The Percentage of Si (%) |
|--------|-----|------|------|------|------|------|------|
| | | C1 | C2 | C3 | O1 | O2 | |
| Ct1 | 0.13 | 71.50 | 24.34 | 4.16 | 61.83 | 28.17 | 15.21 |
| F1 | 0.19 | 73.14 | 23.49 | 3.37 | 79.02 | 20.98 | 6.04 |
| Ct2 | 0.31 | 51.40 | 32.24 | 16.36 | 61.10 | 38.90 | 8.84 |
| F2 | 0.29 | 65.32 | 26.95 | 7.73 | 68.54 | 31.46 | 6.29 |
| Ct3 | 0.32 | 52.70 | 34.93 | 12.37 | 62.01 | 37.99 | 7.77 |
| F3 | 0.31 | 61.76 | 28.41 | 9.83 | 70.32 | 29.78 | 4.95 |

As shown in Table 4, C1 and C2 were the main components of bamboo, to which C1 made a greater contribution. After heat treatment, only C1 increased, while C2 and C3 decreased, indicating that the relative C–C content increased, while the relative contents of C–O, C–OH, C=O, and O–C–O decreased. This was because C1 was derived from lignin, while C2 and C3 were derived from cellulose and hemicellulose, the latter of which degraded at high temperatures. In the O 1s spectra, the relative content of O1 (O–C=O) increased, mainly due to lignin and extracts, while the relative content of O2 (C–O–), which was related to hemicellulose and cellulose, decreased. Crystalline cellulose only decomposes above 240 °C [32], while the temperature in this study was 175–180 °C, so the main component that underwent thermal decomposition was hemicellulose. In the FTIR spectra, the decreased peak intensity at 1730 cm$^{-1}$ indicated the thermal degradation of carbonyl groups in the hemicellulose. Carbonyl groups cleaved and formed carboxyl groups (acetic acid) after heat treatment [30]. Additionally, the increased presence of carboxyl groups in O1 suggested that the main functional group that decreased in C3 was O–C–O. The increase in the C–O bond at 1031 cm$^{-1}$, and the decrease in the C–OH bonds in O2 and the C–OH and C–O–C bonds in C2, indicated that the degradation of C–O in hemicellulose was less pronounced compared to the formation of C–O groups (alcohol and ester). This increased the relative lignin content, resulting in an increase in C1 and O1 peaks. In addition, the relative content of O1 (O–C=O) in the bamboo outer layer board showed the greatest change, which may be an important reason why it showed the darkest color. The pyrolysis of lignin was a relatively complex reaction, and decarboxylation was a part of its condensation reaction at high temperatures. During this process, new chromogenic groups were generated, and the number of chromogenic methoxy groups (O2) decreased, resulting in an overall change in the color of flattened bamboo board [33,34]. Hemicellulose degraded upon processing bamboo into flattened bamboo board, and the brightness of the bamboo also gradually decreased, which indicated that changes in hemicellulose may have been a factor affecting the brightness of the bamboo. Through XPS analysis, it can be observed that the color of bamboo material is not only related to functional groups such as carbonyl, C=C, and C–O as analyzed in FTIR spectra, but is also influenced by the relative content of O–C=O and C–OH. Therefore, in order to gain a deeper understanding of the mechanism behind color variation in bamboo material, it is necessary to combine FTIR and XPS analysis.



In addition to the decreasing Si content along the radial direction of bamboo culm (from the outer to the inner), the relative Si content in the flattened bamboo board was significantly higher than that of the control materials (the bamboo outer layer board vs control material: 15.21% vs. 6.04%; the outer board vs. control material: 8.84% vs. 6.29%; the inner board vs control material: 7.77% vs. 4.95%), which was consistent with previous findings on glossiness. Therefore, the Si content also influenced the glossiness of the flattened bamboo board.

## 4. Conclusions

By comparing and analyzing the visual physical quantities, anatomical structure, and chemical composition of flattened bamboo board and the control material, and their anatomical structure, the following conclusions can be drawn:

The brightness and yellow-blue axis index of flattened bamboo boards decreased after high-temperature softening treatment, while the red-green axis index and glossiness increased. Comparing the SEM images showed that the gradient structure, cell wall proportion, cell cavity size, inclusions, and waxy layer affected the color and glossiness of bamboo material's surface. According to the FTIR spectra, the darkest color of the bamboo outer layer was mainly attributed to the presence of C–H bonds in the wax layer and C=C bonds in the aromatic ring skeleton at 1600 $cm^{-1}$ and 1509 $cm^{-1}$. The outer board was mainly red, possibly due to changes in the C–O content at 1239 $cm^{-1}$. The inner board was mainly yellow, which may have been caused by the C–H stretching vibration in the guaiacyl and syringyl of lignin at 1108 $cm^{-1}$. XPS analysis showed that C1 and O1 increased, while C2, C3, and O2 decreased, indicating that the hemicellulose degraded at high temperatures, which increased the relative lignin content. Changes in the relative content of oxygen-containing functional groups and $SiO_2$ in the flattened bamboo board were important factors responsible for its visual physical quantities change. However, the color of bamboo was greatly influenced by its microstructure and chemical composition. Both the compression and deformation of cells, as well as the changes in the relative content of chromophores, could cause significant changes in color index. The glossiness was also affected by the microstructure and chemical composition, while the impact of chemical composition changes on glossiness may not be as significant as color index. Moreover, changes in the microstructure only lead to a limited decrease in surface roughness, resulting in a less noticeable change in glossiness ($G_{ZL}$ and $G_{ZT}$) compared to color index.

By investigating the influencing factors on the visual characteristics of flattened bamboo boards, it becomes possible to control and process the desired visual characteristics. This enables the formation of standardized materials for the visual characteristics of flattened bamboo boards, thereby facilitating the production of bamboo furniture with superior visual effects.

**Author Contributions:** L.C., C.L. and Z.W. conceived and designed the experiments; C.L. performed the experiments; C.L. and M.C. analyzed the data; L.C. and C.L. wrote the paper. All authors have read and agreed to the published version of the manuscript.

**Funding:** This work was supported by the Guizhou Provincial Basic Research Program (Natural Science) (Grant No. ZK [2021] YB160); the Science and Technology Project of Guizhou Provincial Education Department (Grant No. KY [2021] 112); 13th Five-Year the National Key Research and Development Project (Grant No. 2018YFD0600304); China Postdoctoral Science Foundation (2023M730485).

**Data Availability Statement:** Data available within the article.

**Acknowledgments:** The authors expresses their appreciation to: (1) Nanjing Forestry University and Guizhou Minzu University, for field and laboratory data acquisition support; (2) the insititutions for financial support; (3) three anonymous peer reviewers and the Forests editors for their constructive comments and suggestions, which ultimately yielded an improved contribution.

**Conflicts of Interest:** The authors declare no conflict of interest. The funding sponsors contributed reagents/materials/analysis tools in this paper.

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
