# Peer review of "Effects of Microstructure and Chemical Composition on the Visual Characteristics of Flattened Bamboo Board"

_forests, doi:10.3390/f14112220_

Round 1

Reviewer 1 Report

Comments and Suggestions for Authors

Dear authors please see below for  comments on your manuscript:

The summary is well-written and provides a clear and concise overview of the main objectives, methods, and results of the research on the impact of processing on the visual effect of flattened bamboo panels.

The introduction suggests that research on flattened bamboo wood is primarily focused on technologies and mechanisms for softening and flattening, as well as on the performance and visual characteristics of the surface. However, the factors influencing the visual features of flattened bamboo wood have not been clarified. It also suggests that the authors should expand their references. Here are some studies that describe the problem you are interested in:

- Lou, Z., Wang, Q., Sun, W., Zhao, Y., Wang, X., Liu, X., & Li, Y. (2021). Bamboo flattening technique: a literature and patent review ¹.

- Yuan, T., Han, X., Wu, Y., Hu, S., Wang, X., & Li, Y. (2021). A new approach for fabricating crack-free, flattened bamboo board and the study of its macro-/micro-properties ².

- Zhang, Y., & Li, Y. (2022). Using Statistical Methods to Comparatively Analyze the Visual Characteristics of Flattened Bamboo Boards in Different Bamboo Culms ³.

- Study on the Effect of Flattening Modification on Bamboo Cutting Board .... https://www.mdpi.com/1999-4907/14/4/809.

Materials and Methods

-          Material Preparation Process Specifications: Add more information about the specific steps in the preparation process of flattened bamboo to facilitate experiment reproducibility. This includes details about the time and temperature parameters used.

-          Sample Dimension Details: You can provide additional information about the precise dimensions of the samples, such as tolerances and measurement methods, to enable better interpretation of the results.

-          Use of Control Samples: Consider adding information about the purpose of the control samples and how they were selected. This can help readers better understand the experimental setup.

-          References: If possible, include references that support your choice of material and processes. This can lend additional credibility to your research and substantiate your claims.

-          Visual Material: Introduce a description of the image or graph mentioned in the text (Figure 1) so that readers can better understand your findings.

3. Results and Discussion

3.1. Comparison of Visual Physical Quantities between Flattened Bamboo Boards and Control Samples

-          Visual Observations and Color Analysis: The initial description of the color changes in bamboo due to high-temperature softening treatment is clear and valuable. The inclusion of Figure 1 provides a visual representation of the changes, enhancing the understanding of the results.

-          Quantitative Color Analysis: The transition from visual observations to quantitative color analysis is well executed. The use of brightness (L*), red-green axis (a*), and yellow-blue axis (b*) color indexes is standard practice and aligns with established color science.

-          Interpretation of Color Index Changes: The interpretation of the color index changes is insightful. The explanation of how L*, a*, and b* values changed, and their implications on the overall appearance of the flattened bamboo boards, is clear and well-supported.

-          Comparison between Different Bamboo Boards: The comparison of color changes between the bamboo outer layer board and the outer and inner boards is an essential aspect of the analysis. The substantial ΔE* value (color difference) highlights the significant variation in color after treatment. However, it may be beneficial to discuss potential reasons for these differences in more detail, such as the anatomical structure of the bamboo.

-          Comparison with Prior Research: The reference to previous studies on wood color changes under heat treatment (citing [13], [14], and [15]) adds depth to the discussion and places the findings in the context of existing scientific knowledge.

-          Clarification of L, a, and b* Trends:** To further enhance the discussion, consider explaining why L* and b* decreased while a* increased after high-temperature treatment. Discuss possible mechanisms or chemical reactions that might be responsible for these trends, adding a deeper layer of understanding to the results.

3.2. Effects of Microstructure Differences on Visual Physical Quantities

-          Microstructural Changes and Impact on Color and Glossiness: Your analysis of microstructural changes in bamboo is detailed, but it would be useful to highlight specific quantitative data or analytical methods supporting the claims regarding changes in color and glossiness. For example, how were specific color and gloss parameters measured?

-          Glossiness and Microstructure: Connect the microstructure data with the gloss results. Consider how factors such as cavity size and other microstructural features influence changes in surface glossiness. This can further enhance the understanding of the mechanism behind changes in gloss.

-          More Detailed Microstructural Analysis: In addition to describing changes in cell and cell wall shapes, consider conducting additional analysis of microstructure in the context of compression and deformation. How are these changes related to the optical properties of the material?

-          Impact of Inclusions on Gloss: Explain the impact of inclusions on the surface gloss of bamboo. Discuss how the presence of inclusions on the surface of the outer layer affects light reflection and gloss.

-          Discussion of Innovation: If your results are innovative or differ from previous research, emphasize this innovation in your analysis. How does your research contribute to the understanding of color and gloss changes in bamboo at the microstructural level?

The conclusion is well-structured and clear. However,while the conclusion is scientifically sound, you can to include a brief section outlining potential practical implications of the findings. How might these results be applied in industries or fields that use bamboo-based materials?

Author Response

Thank you so much for your careful reading and very helpful comments, corrections.We tried our best to revise the manuscript according to your comments and the manuscript has been improved much (Red). Thank you very much for your time and helping us revise the manuscript and making it look better. Please see the attachment for the modification.

  1. It also suggests that the authors should expand their references. Here are some studies that describe the problem you are interested in:

Authors’ response: Thank you very much for your suggestion. It was added in the revised manuscript.  

-“Zhao, Y.; Ma, Y.; Lou, Z.; Li, Y. Study on the Effect of Flattening Modification on Bamboo Cutting Board and Corresponding Carbon Footprint Evaluation. Forests 2023, 14, 809.”-This article was not cited in the revised manuscript due to an editorial request to reduce citations to Forests journal papers.

  1. Material Preparation Process Specifications: Add more information about the specific steps in the preparation process of flattened bamboo to facilitate experiment reproducibility. This includes details about the time and temperature parameters used.

Authors’ response: Thank you very much for your suggestion. We have add more information in the revised manuscript. (Page 2, line 80~81)

  1. Sample Dimension Details: You can provide additional information about the precise dimensions of the samples, such as tolerances and measurement methods, to enable better interpretation of the results.

Authors’ response: Thank you very much for your suggestion. We have add more information in the revised manuscript (Page 2, line 80-81). However, the precise dimensions of the samples has no effect on the determination of color and gloss parameters, so it is not given in the manuscript.

  1. Use of Control Samples: Consider adding information about the purpose of the control samples and how they were selected. This can help readers better understand the experimental setup.

Authors’ response: Thank you very much for your suggestion. The information had been revised and added in the revised manuscript.(Page 2, line 84-91)

5.References: If possible, include references that support your choice of material and processes. This can lend additional credibility to your research and substantiate your claims.

Authors’ response: Thank you very much for your suggestion. Some additions have been made to the literature as you suggested.

6.Visual Material: Introduce a description of the image or graph mentioned in the text (Figure 1) so that readers can better understand your findings.

Authors’ response: Thank you very much for your suggestion. The description of the figure 1 had been introduced in the part of materials and the graph legend, so we didn’t give the additional description.

7.Visual Observations and Color Analysis: The initial description of the color changes in bamboo due to high-temperature softening treatment is clear and valuable. The inclusion of Figure 1 provides a visual representation of the changes, enhancing the understanding of the results.

Authors’ response: Thank you very much for your recognition.

  1. Quantitative Color Analysis: The transition from visual observations to quantitative color analysis is well executed. The use of brightness (L*), red-green axis (a*), and yellow-blue axis (b*) color indexes is standard practice and aligns with established color science.

Authors’ response: Thank you very much for your recognition.

9.Interpretation of Color Index Changes: The interpretation of the color index changes is insightful. The explanation of how L*, a*, and b* values changed, and their implications on the overall appearance of the flattened bamboo boards, is clear and well-supported.

Authors’ response: Thank you very much for your recognition.

10.Comparison between Different Bamboo Boards: The comparison of color changes between the bamboo outer layer board and the outer and inner boards is an essential aspect of the analysis. The substantial ΔE* value (color difference) highlights the significant variation in color after treatment. However, it may be beneficial to discuss potential reasons for these differences in more detail, such as the anatomical structure of the bamboo.

Authors’ response: Thank you very much for your suggestion. Some supplements have been revised, and we hope to meet you requirements. (Page 6, line 207-214)

11.Comparison with Prior Research: The reference to previous studies on wood color changes under heat treatment (citing [13], [14], and [15]) adds depth to the discussion and places the findings in the context of existing scientific knowledge.

Authors’ response: Thank you very much for your recognition.

12.Clarification of L, a, and b* Trends:** To further enhance the discussion, consider explaining why L* and b* decreased while a* increased after high-temperature treatment. Discuss possible mechanisms or chemical reactions that might be responsible for these trends, adding a deeper layer of understanding to the results.

Authors’ response: Thank you very much for your suggestion. The discuss was added in the revised manuscript. (Page 9, line 311-316)

  1. Microstructural Changes and Impact on Color and Glossiness: Your analysis of microstructural changes in bamboo is detailed, but it would be useful to highlight specific quantitative data or analytical methods supporting the claims regarding changes in color and glossiness. For example, how were specific color and gloss parameters measured?

Authors’ response: Thank you very much for your suggestion. This suggestion is very good, and we will conduct detailed research on the relationship between structure quantification and color or glossiness quantification based on this suggestion in the future study.

14.Glossiness and Microstructure: Connect the microstructure data with the gloss results. Consider how factors such as cavity size and other microstructural features influence changes in surface glossiness. This can further enhance the understanding of the mechanism behind changes in gloss.

Authors’ response: Thank you very much for your suggestion. The discuss was added in the revised manuscript. (Page 7, line 233-235; line 240-243)

15.More Detailed Microstructural Analysis: In addition to describing changes in cell and cell wall shapes, consider conducting additional analysis of microstructure in the context of compression and deformation. How are these changes related to the optical properties of the material?

Authors’ response: Thank you very much for your suggestion. The relationship between micro-structure changes and the optical properties of material was added in the revised manuscript. (Page 7, line 240-243)

16.Impact of Inclusions on Gloss: Explain the impact of inclusions on the surface gloss of bamboo. Discuss how the presence of inclusions on the surface of the outer layer affects light reflection and gloss.

Authors’ response: Thank you very much for your suggestion. The discuss was added in the revised manuscript. (Page 7, line 233-235)

17.Discussion of Innovation: If your results are innovative or differ from previous research, emphasize this innovation in your analysis. How does your research contribute to the understanding of color and gloss changes in bamboo at the microstructural level?

Authors’ response: Thank you very much for your suggestion. We have explained the changes in color and gloss due to structural differences from the aspects of micro-structure and chemical composition, which can be a basis for future quantitative research on the correlation between structural and visual characteristics, and then guide the processing technology of bamboo flattened furniture surface decoration materials.

18.The conclusion is well-structured and clear. However,while the conclusion is scientifically sound, you can to include a brief section outlining potential practical implications of the findings. How might these results be applied in industries or fields that use bamboo-based materials?

Authors’ response: Thank you very much for your suggestion. The conclusion was added this section in the revised manuscript, and hope to meet your requirement. (Page 13, line 443-455)

Reviewer 2 Report

Comments and Suggestions for Authors

The abbreviations Ag and C in Table 1 are not explained in the text and what these abbreviations mean should be included in the text.

It should be stated whether the cracking process in the cells is due to heat treatment or flattening and what type of press was used in this flattening process. The percentage of thickness loss in the samples during the smoothing process should be given.

In later studies, the authors can examine the change in the amount of C, H, O , Si, and other important minerals with SEM EDX, but it does not allow you to make a detailed analysis as in XPS analysis.

Author Response

Thank you so much for your careful reading and very helpful comments, corrections.Thank you very much for your time and helping us revise the manuscript and making it look better. Please see the attachment for the modification.

1.The abbreviations Ag and C in Table 1 are not explained in the text and what these abbreviations mean should be included in the text.

Authors’ response: Thank you very much for your suggestion. We have deleted them from the revised manuscript.

  1. It should be stated whether the cracking process in the cells is due to heat treatment or flattening and what type of press was used in this flattening process. The percentage of thickness loss in the samples during the smoothing process should be given.

Authors’ response: Thank you very much for your suggestion. Since the preparation process of bamboo flattened boards was described in our previous article [10], it has been indicated in this manuscript that the bamboo flattened boards was prepared by referring to the previous article, so these parameters were not listed in this manuscript.

3.In later studies, the authors can examine the change in the amount of C, H, O, Si, and other important minerals with SEM EDX, but it dose not allow you to make a detailed analysis as in XPS analysis.

Authors’ response: Thank you very much for your suggestion. In our later studies, we will consider to examine the change in the amount of C, H, O, Si, and other important minerals with SEM EDX, so as to more intuitively analyze the correlation between structure and chemical composition.

Reviewer 3 Report

Comments and Suggestions for Authors

The manuscript entitled "Effects of Microstructure and Chemical Composition on the Visual Characteristics of Flattened Bamboo Board" is interesting. However, some things need to be explained in more detail:

It is necessary to explain further how to differentiate Ct1, Ct2, and Ct3 in sample selection. Likewise, it is necessary to explain how samples are taken for F1, F2, and F3.

In SEM analysis, it is necessary to explain the three analysis groups in question.

There needs to be an extension for FTIR and  XPS; and it needs to be explained in more detail about the differences in information to be obtained from the two analyses.

Statistical analysis does not yet explain the design of the experimental analysis.

The calorimeter analysis results for visual physical quantities do not yet explain why the parallel to the processional texture (GZL) and perpendicular to the texture (GZT) values are lower than the other results

How do you explain that glossiness is related to surface flatness when you did not test for surface flatness in the study (line 206)

It is necessary to mark the highlights that you want to describe in the SEM analysis results image (Figure 4).

What magnification is used for Figure 4

It is necessary to explain better the use of the terms outer layer board, outer board and inner board. It would be better if there were illustrations

Conclusions need to be reformulated so that the results do not become a repetition. The conclusion should convey a focus on the important findings obtained

Comments on the Quality of English Language

THe English should be improved

Author Response

Thank you so much for your careful reading and very helpful comments.Thank you very much for your time and helping us revise the manuscript and making it look better. Please see the attachment for the modification.

  1. It is necessary to explain further how to differentiate Ct1, Ct2, and Ct3 in sample selection. Likewise, it is necessary to explain how samples are taken for F1, F2, and F3.

Authors’ response: Thank you very much for your suggestion. We have added in the revised manuscript. (Page 2, line 73-75;Page 7, line 80-91)

  1. In SEM analysis, it is necessary to explain the three analysis groups in question.

Authors’ response: Thank you very much for your suggestion. We have added in the revised manuscript. (Page 6, line 207-214;Page 7, line 233-235; 240-243) 

  1. There needs to be an extension for FTIR and XPS; and it needs to be explained in more detail about the differences in information to be obtained from the two analyses.

Authors’ response: Thank you very much for your suggestion. We have added in the revised manuscript. (Page 10, line 347-350;Page 12, line 394-402;412-417)

  1. Statistical analysis does not yet explain the design of the experimental analysis.

Authors’ response:  Figures 2,5, 6, 7, and 8 were analyzed and processed by Origin, so we believe there are no issues with the descriptions in the statistical analysis. We hope our response can satisfy you.

  1. The calorimeter analysis results for visual physical quantities do not yet explain why the parallel to the processional texture (GZL) and perpendicular to the texture (GZT) values are lower than the other results

Authors’ response:  We have add more information in the revised manuscript. (Page 7, line 240-243;Page 13, line 443--450 ).

  1. How do you explain that glossiness is related to surface flatness when you did not test for surface flatness in the study (line 206)

Authors’ response: Thank you very much for your suggestion. We have add more information in the revised manuscript. (Page 3, line 113-114, Page 7,line 229 ).

Glossiness is a crucial parameter reflecting the ability to reflect light on the surface of a material, which is correlated with factors such as the observation environment, material roughness, and molecular structure. The relationship between the incident light intensity and the reflected light intensity of the mirror surface of the object is as follows:

Where I0 is the intensity of incident light; I is the intensity of the specular reflection light; θ is the angle of incidence; γ is the wavelength of light;ρ represents surface roughness.According to this formula, it can be found that as the surface roughness decreases, the value of I increases.

  1. It is necessary to mark the highlights that you want to describe in the SEM analysis results image (Figure 4).

Authors’ response: Thank you very much for your suggestion. We have made modifications in Figure 4.

  1. What magnification is used for Figure 4

Authors’ response: Thank you very much for your suggestion. We have made modifications in Figure 4.

  1. It is necessary to explain better the use of the terms outer layer board, outer board and inner board. It would be better if there were illustrations

Authors’ response: Thank you very much for your suggestion. We have added in the revised manuscript. (Page 2, line 80-81)

  1. Conclusions need to be reformulated so that the results do not become a repetition. The conclusion should convey a focus on the important findings obtained.

Authors’ response: We have add more information in the revised manuscript. (Page 13, line 429--455 ).

Round 2

Reviewer 1 Report

Comments and Suggestions for Authors

Dear authors,

I would like to express my gratitude for your prompt and meticulous work on the revisions in your manuscript. I have received your message and reviewed the changes you have made based on my comments. I am pleased to say that I am satisfied with the improvements you have implemented.

Your efforts in adding references, expanding information about material preparation, and providing more details about the samples have been noticed and appreciated. I am also glad to see that you have added additional information about the use of control samples and updated the literature to better support your research. The manuscript now looks much improved and of higher quality.